# The Economics of Rabbit Farming: A Pilot Study on the Impact of Different Housing Systems

**DOI:** 10.3390/ani11113040

**Published:** 2021-10-23

**Authors:** Chiara Mondin, Samuele Trestini, Angela Trocino, Guido Di Martino

**Affiliations:** 1Department of Land, Environment, Agriculture and Forestry (TESAF), University of Padova, 35020 Legnaro, Padova, Italy; chiara.mondin@unipd.it; 2Department of Comparative Biomedicine and Food Science (BCA), University of Padova, 35020 Legnaro, Padova, Italy; angela.trocino@unipd.it; 3Istituto Zooprofilattico Sperimentale delle Venezie, 35020 Legnaro, Padova, Italy; gdimartino@izsvenezie.it

**Keywords:** bicellular cage, dual purpose cage, enriched cage, production costs, animal welfare

## Abstract

**Simple Summary:**

The welfare of farmed rabbits is a growing concern for consumers, who are demanding animal-friendly housing systems. At the same time, it is important to understand the impact of such a system on the economic sustainability of this sector. Through a face-to-face interview, we collected data of structures, productivity, and costs from six farms using conventional or structurally enriched cages. Looking at the results, the enriched cage, in comparison with conventional housing systems, is both non-penalizing and economically sustainable. In addition, the adoption of a more animal-friendly housing system leads to a reduction of drugs cost. This result is opening the discussion on the opportunity to contribute to the reduction of antibiotic use, therefore pursuing an improvement of animal welfare. Due to the complexity of the topic, we believe that further studies on the economic sustainability of this sector are needed to confirm our results.

**Abstract:**

This research evaluates the economic sustainability of rabbit farms using different housing systems—bicellular (BI), conventional dual-purpose (DP) and enriched cages designed according to the World Rabbit Science Association guidelines (WRSA)—through a field-based study involving six farms over the course of five years. The cages were compared based on three productivity indices expressed in kg of produced live weight/m^2^ and on eight cost indices expressed in EUR/kg of produced live weight. The results showed that WRSA significantly reduced the productivity index per walkable cage area in buildings and cages, thanks to the longer platform area included in the cage compared to the other systems. Concerning cost indexes, total variable costs were not different among housing systems, whereas significant differences were observed within costs items. As for the feed costs, DP underperforms compared to BI or WRSA (1.15 vs. 1.02 and 0.99 EUR/kg produced live weight); for drugs costs, BI was less competitive compared to DP and WRSA (0.12 vs. 0.06 and 0.05 EUR /kg). In conclusion, under the conditions of the present study, the economic results of farms that adopted housing systems designed to improve rabbit welfare, such as WRSA enriched systems, were economically sustainable and, comparable to conventional housing systems based on BI or DP cages, also provided a significant reduction in drug use in the tested farms. A comprehensive collection of data from more farms at a European level would be necessary to confirm these results on the economics of farms adopting alternative housing systems for rabbits.

## 1. Introduction

During the last 15 years, consumers’ concerns about rabbit breeding practices have grown [1,2]. That’s why, over the years, the importance of good practices has been studied, focusing on performance, behaviour, and the effect of housing conditions (impact of flooring, environmental enrichment, available surface, and social isolation) on rabbits [3,4,5,6,7,8]. Nowadays, the Farm to Fork strategy [9] of the European Green Deal [10] calls for new regulations on animal welfare and management, with a special focus on housing systems and animal health, aiming at reducing the use of antimicrobials (AMU) for farmed animals by 50% by 2030, which responds to consumers’ concerns [11]. In this regard, limited data are available on AMU in farmed rabbits, which was found to be much higher compared to other livestock [12]. According to an Italian study on rabbit farmers’ perceptions, most of interviewed people thought that AMU can be decreased to 20–30% [13].

No specific EU rules yet exist for farmed rabbits, but the European Parliament resolution of 14 March 2017 [14] on minimum standards for the protection of farm rabbits calls for alternatives in housing and management of farmed rabbits, which is further stressed in the European Parliament Resolution on the European Citizens’ Initiative ‘End the cage age’ [15].

Currently, as for housing systems, conventional farms use conventional cages (bicellular cages for growing rabbits and dual-purpose cages for both reproducing does and growing rabbits) or structurally enriched cages designed according to guidelines of the World Rabbit Science Association (WRSA) [8]. Both have an automatic system for the distribution of water (usually through a nipple system) and feed (complete pelleted diets are usually used). Most of the commercial production comes from hybrids (commercial crossbred rabbits) and artificial insemination is largely adopted. The replacement rate of reproducing does can reach 120% per year [8,16].

Farms equipped with bicellular cages (BI) house reproducing does with the offspring until weaning (30–35 days of age) when kits are moved to bicellular cages (1–2 animals/cage) for fattening and until slaughter. In the case of dual-purpose cage (DP), at the time of weaning, the doe is moved to a clean and disinfected enclosure and the kits remain in their cage until the end of fattening (all-in/all-out system). The cage is equipped with a removable nest box for offspring and lactation, which is taken away for the fattening period. Structurally enriched cages, also called welfare-cages or WRSA, are of recent use. The management of animals in WRSA cages works the same as in dual-purpose cages, but cages are higher and with enrichments (a raised platform and a plastic footrest) providing more surface area for the fattening rabbits and an opportunity for movement [8]. For more information about the production and housing systems for farmed rabbits, refer to the latest Scientific Opinion of the European Food and Safety Authority [8,17]. This Scientific Opinion [8] also compared the health and welfare of rabbits kept under different production systems by an overall impact score based on both health- and behaviour-related welfare consequences and obtained through an Expert Knowledge Elicitation (EKE) process. According to the EKE results [8], welfare and health of reproducing does is likely lower in conventional cages compared to other systems (among which enriched cages and parks). As for growing rabbits, welfare is highly likely to be lower in conventional cages (BI and DP cages) and higher in indoor parks, whereas no distinction can be made among the WRSA cages and niche systems (litter parks, outdoor systems, organic systems) [8].

One of the greatest limitations for farmers in changing the housing system and adopting the best technology depends on individual choices and preferences, often linked to economic advantages. Farmers can choose to adopt higher standards in animal welfare if they also gain advantages in terms of a higher price paid by the market (dictated by one or both of the following: higher quality of meat or higher consumer willingness to pay) or in terms of a cost reduction [18].

However, market dynamics influence the economic performance of the sector, as farmers are also subjected to high price volatility over the past 10 years, due to a cyclical market crisis within a trend of decline in consumption [19,20,21,22]. Variable costs range from 62% to 93% of the total revenue and feeding costs at 50–60% of variable costs, and the instability of market prices can often undermine the economic sustainability of the business [16,23].

The possibility of farmers to adopt any animal welfare innovation following EU regulations largely depends on the impact of such innovation on the economics of the business. This is an urgent issue in Europe and Italy, the third largest EU producer, which, together with Spain and France, supplied 83% of the total EU production of professional rabbit farming in 2018 [21,24]. Within Italy, the north-eastern regions of the country are the leaders in terms of rabbit meat production (66% of national production, almost 11 million slaughtered animals) [24,25].

The literature cited above has addressed the issue of cost analysis by adopting the approach of the typical farm. This method does not require the collection of data at the farm-level for a significant number of farmers. As such, they are a considerably lower cost option relative to farm surveys, but they do not generally ensure the representativeness of the results [26].

To contribute to the debate on the economic advantages and disadvantages related to the adoption of animal-friendly housing systems in rabbit farms, this study aimed at comparing the economic performance of farms adopting three housing systems for reproducing does and growing rabbits, i.e., conventional cages for does and bicellular cages for growing rabbits (BI), and dual-purpose cages (DP) or structurally enriched cages (WRSA) for both categories, based on farm-level survey. To the best of our knowledge, this is the first attempt at performing a statistical analysis that compares, on the basis of the main cost elements observable at farm-level, these three housing systems for rabbit farms.

## 2. Materials and Methods

### 2.1. Data Collection

Face to face interviews were conducted on 6 farms between February and April 2019 and information was collected over the 5-year period between 2014–2018, regarding the type of rabbit housing, productivity and costs. Farms were located in the Regions of Veneto and Friuli Venezia Giulia, north-east Italy. They were selected based on the housing systems adopted for growing rabbits: bicellular (BI), conventional dual-purpose (DP), and structurally enriched cages (WRSA). A total number of 19 observations (i.e., data referring to the yearly production) were collected from six farms (identified as A, B, C, D, E, and F). Data relating to periods of less than one year were excluded from the analysis. One of the farms (farm C) changed the housing system in 2017, from bicellular to WRSA. A different number of observations was available per farm, for which an unbalanced dataset was used with 9 observations from farms with bicellular cages (BI) (9 observations × 3 farms); 7 observations from farms with conventional dual-purpose cages (DP) (7 observations × 2 farms); and 3 observations from farms with structurally enriched cages (WRSA) (3 observations × 2 farms). Regarding the evaluation of the age of the cages, this was possible for DP, installed on the 2 farms in 2010 and 2015, and for WRSA, installed on the 2 farms in 2017. BI cages were installed before 2014. The production systems were consistent with the description provided by EFSA [8] and all specific characteristics of the six farms are given in Table 1.

### 2.2. Cost Calculation

All costs incurred, both variable costs (VC) and fixed costs (FC) were attributed to the final product, i.e., kilogram of produced live weight. Among VC, the analysis considered the following: feed, drugs (including veterinary service costs), reproduction (costs of replaced does and artificial insemination), and energy costs (these costs are assumed to be those more affected by the housing system and easy to analytically determine). Feed costs included all the quantity of feeds (medicated or not) consumed in one year on the farm for all categories of rabbits [27]. All costs incurred were obtained based on the bills of each year of observation per farm. Since farmers in the sample were usually organised as a family farm with a diversified production, the labour costs associated with rabbit breeding was difficult to determine analytically and, thus, labour costs were not considered in this study. Additionally, costs for insurances, social security, disposal of carcasses, feces and wastes were not considered because they were not analytically identified in the farm account. FC were estimated from the value of the structures (shed + plants + cages) and based on commercial proposals retrieved from a leading company in the sector. Costs were expressed per cage and per square meter, in the case of a shed. The estimated purchasing cost of a shed was equal to EUR 448/m^2^, including masonry works, automatic feeding system (with silos), cleaning system (scrapers), air conditioning system, and electrical and water system. The cost of the cages changed according to the housing system. The price of WRSA and DP cages was EUR 78.08 and EUR 70.15 and included 1 fattening/doe cage plus half a replacement/recovery cage. For BI, the price was EUR 18.30 for the fattening cage and EUR 47.77 for the doe cage. Fixed costs were determined as depreciation costs considering a depreciation time of 20 years for the structures (shed plus services) and 10 years for the cages (Table 2).

To allow a comparison among different years, all the monetary values were discounted to 2018 based on the price index of the goods purchased by farmers [28]. As VAT is both a return and a cost in the considered type of farms, the VAT was considered as a cost in the analysis.

### 2.3. Data Analysis

The performance of the farms using the different housing systems in the present survey was measured using both technical and economic indexes. The technical indexes give productivity information and are expressed in terms of kilograms of live weight produced in one year per square meter of investment (shed or cage) (kg/m^2^). In detail:Yearly productivity of the walkable cage area (kg/m^2^): this was obtained by dividing the produced live weight in one year by the area (m^2^) of cages available to rabbits. This is equal to the cage floor for BI and DP cages, and the cage floor plus platform for WRSA cages. This index includes the areas of both fattening and reproduction cages.Productivity of the base cage area (kg/m^2^): this was obtained by dividing the slaughtered live weight in one year by the area (m^2^) of the cage, which excludes the platform area in WRSA. This index includes the areas of both fattening and reproduction cages.Productivity of the total shed area (kg/m^2^): this was obtained by dividing the slaughtered live weight in the year by the area (m^2^) of the shed declared by farmers. In the all-in/all-out system, all shed areas are considered in the calculation.

The economic indexes give the average monetary costs of different cost items per kilogram of live weight produced in one year (EUR/kg) (Table 3), calculated as a ratio between the total yearly cost item divided by the total live weight production in the year:Cost indexes (EUR/kg): these were obtained by dividing the yearly expenditure for each cost items by the produced live weight in one year, expressed in kg. “Total variable costs” (VC) includes feed, drugs, energy, and reproduction. The economic information comes from the farm accountancy, the data recorded by the breeders, or both. “Total fixed costs” (FC) includes the depreciation costs of the shed and cage.

The analysis of differences in productivity and economic indexes among housing systems started verifying the ANOVA assumptions: the normality distribution of the residuals (Shapiro-Wilk test) and homoscedasticity (White test, Breusch-Pagan test, and F-test). All the variables have a normal distribution and were analysed using a two-way ANOVA, taking the fixed effect of the housing system and management. When heteroscedastic distribution was detected (costs for feed and reproduction), the Welch ANOVA was performed. The Tukey-Kramer test and Games-Howell post-hoc tests were performed.

To assess the existence of a farm effect, the one-way ANOVA between-subjects was performed, looking for statistically significant effects associated with the management (random effects).

The differences between the means with a *p*-value < 0.05 were accepted as statistically significant. Significance levels of *p*-value < 0.10 were also discussed. The analysis was performed using the IBM SPSS (version 26.0).

## 3. Results

### 3.1. Effect of the Housing System

Table 3 presents an insight into the productivity of farms with different housing systems.

The yearly productivity of the walkable cage area was lower in farms with WRSA cages (158.08 kg produced live weight/m^2^) compared to those with DP cages (196.82 kg/m^2^) (*p*-value < 0.01). There is also a trend towards a significant effect (*p*-value = 0.07) between WRSA and BI, with a higher value in the latter.

The productivity of the base cage area (kg produced live weight/m^2^) did not differ among housing systems, despite the fact that the productivity of the total shed area tended to be higher in farms using BI cages (115.94 kg/m^2^) compared to those using DP cages (100.26 kg/m^2^) (*p*-value = 0.07), which could be attributed to the second floor of BI cages, compared to the flat-deck DP cages.

As for differences in the cost indexes (Table 3), VC, as a whole, did not differ among farms with different housing systems, whereas the cost of the single items showed some differences. In detail, feed costs in farms using BI and WRSA housing systems (1.02 and 0.99 EUR/kg produced live weight, respectively) were significantly lower than what was recorded in farms with DP cages (EUR 1.15/kg) (*p*-value < 0.01). The drug cost was significantly higher in farms adopting BI housing (EUR 0.11/kg produced live weight) compared to those with DP (EUR 0.06/kg) and WRSA cages (EUR 0.05/kg) (*p*-value < 0.01). The energy costs were higher in farms with BI and DP cages (0.06 and 0.07 EUR/kg produced live weight, respectively) than in those with WRSA cages (EUR 0.03/kg) (*p*-value < 0.01). Concerning reproduction, costs results were statistically higher in farms with WRSA cages (0.19 €/kg) than in those with BI (0.08 €/kg), and tended to be higher than DP cages (EUR 0.08/kg) (*p*-value = 0.08).

Regarding FC, despite the significance of the F-test, no difference among farms with different housing systems was recorded. Nevertheless, the shed cost tended to be lower for the BI (EUR 0.20/kg slaughtered live weight) compared to DP (EUR 0.23/kg) (*p*-value = 0.08). Finally, the cage cost was significantly higher in farms with WRSA cages (EUR 0.10/kg) compared to those with BI and DP (0.08 and 0.07 EUR/kg, respectively) (*p*-value < 0.01).

### 3.2. Farm Effect

Table 4 presents the estimations of the effects of the farms within the different housing systems, with farms identified as A, B and C using BI cages; farms D and E using DP cages; and farms C and F using WRSA cages.

As for technical indexes, productivity per walkable cage area, base cage area and total shed area significantly changed among farms using BI systems. As for economic indexes, estimations about VC show a significant difference in feed cost among farms within all the housing systems. A significant difference also appears for drugs costs and energy costs among farmers using BI systems and for reproduction costs among farmers using DP or WRSA cages. As for FC, the effect of the farm was significant for shed costs within BI systems and for cage costs within DP cages.

## 4. Discussion

Most of the research about rabbit farming economics capitalizes the productive doe as an economic unit, with all costs, returns and profits reported as doe per year [16,23,27,29], whereas this study assessed the economic impact of the housing system on rabbit live weight production. Thus, direct comparison with previously published results about the economics of rabbit farms was not possible and some data retrieved from the field were used for comparison.

Importantly, despite the limited number of farms included in the survey, the results of the present study about the farm effect on technical and economical indexes underline the key role of the whole production system on the performance of the farm. Production systems are not standardized and account for several differences among farms in rabbit management and housing, as well resumed by EFSA [8], as for biosecurity (e.g., cleaning, disinfection, vaccines, isolation, and duo system, etc.), ambient conditions (indoor, semi-plein air, and outdoor), reproduction (insemination, rhythm, and batch), nutrition and feeding (ad libitum, restricted, complete diets, and forages) and genetics, besides housing (enclosure, group, floor, and enrichment, etc.). Within the different production systems, farming global results differ and, thus, costs and revenues change with the reproductive efficiency of the does, the rate at which meat rabbits reach slaughtering weight at different ages, as well as the morbidity and mortality of the different categories of rabbits.

On the other hand, to the best of our knowledge, this is the first study providing data about how the different housing systems can affect the economic sustainability of farmers in a context characterized by a high variability of all other production factors, which are covered by the evaluation of the meat produced and the variable costs incurred over one year.

We reported that the changes of the housing system, which addressed animal welfare through the use of WRSA cages, did not reduce the yearly productivity of farms in terms of produced live weight per m^2^ base cage area or m^2^ shed area compared to farms using BI and DP systems. Differences in terms of walkable cage area were due to the higher available surface in WRSA cages, thanks to the platforms, which means that WRSA cages provide more movement opportunity (more walkable surface) to animals compared to BI and DP cages, without affecting production per cage area.

As for costs, the results of the present study do not identify significant differences for aggregated costs (variable and fixed), which could be due to the small sample size, the variability of the production factors other than housing among the farms, or a combination thereof. The total costs ranged from 1.54 to 1.66 to 1.54 EUR /kg of live weight produced in farms with BI, D and WRSA housing systems. On average, variable costs accounted for 81.6% of the total, with feed costs being the main item at 66.7% of the total, drugs at 4.6%, energy at 3.4% and reproduction at 7.4%. Fixed costs were 18.4%, with shed costs accounting for 13.1% and cages costs for 5.3% of the total. Field data dating back to 2014 gave a production cost ranging from 2.06 to 1.91 EUR/kg of meat (including labour, insurance, social providence and other costs, which accounted for a 17.1% of the total costs on average) [30].

Under our conditions, farms using WRSA cages were competitive for costs related to feeding, drugs and energy, whereas the worst performance was recorded in farms with BI cages for drug costs and in those with DP cages for feed costs. The analysis of the indexes related to structures (cages) shows that BI and DP appear to have lower depreciation costs than WRSA, mainly due to the lower cost for m^2^ of the cage.

Indeed, feed costs can vary largely, according to all factors affecting feed intakes, such as diet composition (especially dietary energy and fibre contents) and feeding regime (restricted or ad libitum feeding), besides the health of the animals on a farm [27] and, last but not least, the feed company. On the other hand, the reduction of drug costs (i.e., use) could be associated with an improvement of the health and welfare of animals in those farms that adopted alternative housing systems, such as WRSA cages, and adopted the most effective biosecurity practices. Thus, this result in farms with WRSA cages should not be underestimated from a public health point of view; the reduction of drugs in animal production may contribute to control antibiotic resistance, in coherence with EU health policy [8]. Indeed, the drug costs were higher in farms adopting BI, with relevant differences among farmers, which were associated with differences in management, besides the age of the farms (figures were lower in farms with WRSA cages compared to the other housing systems). In fact, farms using WRSA cages underwent a full and complete all-in/all-out for renovation and renewed their reproducing does flock, which usually implies the better health of animals.

As for farms with WRSA cages, the advantages in terms of lower feed, drugs and energy costs are cancelled due to higher costs for reproduction and the depreciation of cages. The former reason was mainly linked to a farm effect within the surveyed farms, whereas the latter depends on the higher purchasing price of WRSA cages compared to the conventional cages, which is a general condition for all farms.

In conclusion, based on these preliminary results, since total FC or VC did not change among housing systems, whereas drug costs were lower in WRSA cages compared to BI cages, we can state that in the perspective of evolution towards good welfare practices in rabbit farming, the replacement of traditional cages with structurally enriched cages, potentially associated with better hygiene conditions, could be economically sustainable, provided that market prices remain acceptable. These results should be confirmed on a larger number of observations. Furthermore, an exhaustive assessment of production costs for the different housing systems should also include labour costs. To reduce the farmers’ effect on performance indexes and enlarge perspectives about alternative housing systems for rabbits, further research should increase the number of farms and include other housing systems, such as elevated parks for conventional farms and organic systems per niche farms.

## Figures and Tables

**Table 1 animals-11-03040-t001:** Characteristics of the housing systems present in the farms surveyed in this study.

Housing System	Conventional Cages for Does and Bicellular Cages for Growing Rabbits (BI)	Conventional Dual-Purpose Cages (DP)	Structurally Enriched Cages (WRSA)
Farms, number	3	2	2
Observations, number	9	7	3
Group-housing system	Growing rabbits	Growing rabbits Does with kits	Growing rabbits Does with kits
Available surface (m^2^/animal) ^a^	0.105–0.167	0.296–0.331 ^b^	0.452–0.508 ^b,c^
Expected ^d^	Good level of feeding, health, and biosecurity measures	More available space compared to BI, group housing of fattening rabbits favouring natural behaviour	Standard cages with plastic/wire-mesh platform, improved animal welfare and allows biosecurity measures

^a^ Area of all cages on the farm; ^b^ The nest area is included; ^c^ The platform surface is included; ^d^ Reference: EFSA [8].

**Table 2 animals-11-03040-t002:** Estimated purchasing costs for different type of cage and structures.

Item		Purchasing Cost ^a^
Housing system		EUR/cage
BI	Cages for rearing rabbits	18.30
	Cages for reproducing does with kits	44.77
DP	Cages for reproducing does with kits and growing rabbits	70.15
WRSA	Cages for reproducing does with kits and growing rabbits	78.08
Structures ^b^		EUR/m^2^
	Automatic feeding system (with silos)	24.40
	Cleaning system (scrapers)	15.70
	Masonry works	318.10
	Air conditioning system	41.85
	Electrical and water system	47.95
	Total	448.00

^a^ Values based on commercial proposals retrieved from a leading company in the sector; ^b^ Hypothesis of a structure with 2 sheds of 42 × 15 metres each (630 m^2^ each and 1260 m^2^ in total).

**Table 3 animals-11-03040-t003:** Estimations of the productivity and economic indices in the surveyed farms using different housing systems (SD in brackets).

Housing System	Bicellular (BI)	DP	WRSA
Observations (number)	9	7	3
Slaughter weight (kg)	2.81 (0.08)	2.71 (0.07)	2.66 (0.05)
Walkable cage area (kg/m^2^)	193.47ab (0.08)	196.82a (0.07)	158.08b (0.05)
Base cage area (kg/m^2^)	193.47a (36.76)	196.82a (27.46)	180.63a (27.62)
Total shed area (kg/m^2^)	115.94a (36.76)	100.26a (27.46)	118.69a (20.20)
Total variable costs (EUR/kg)	1.26a (0.13)	1.36a (0.14)	1.25a (0.12)
Feed (EUR/kg) ^1^	1.02a (0.06)	1.15b (0.17)	0.99a (0.09)
Drugs (EUR/kg)	0.11a (0.05)	0.06b (0.02)	0.05b (0.03)
Energy (EUR/kg)	0.06a (0.03)	0.07a (0.01)	0.03b (0.01)
Reproduction (EUR/kg) ^1^	0.08a (0.06)	0.08ab (0.08)	0.19b (0.14)
Total fixed costs (EUR/kg)	0.28a (0.04)	0.30a (0.04)	0.29a (0.14)
Shed (EUR/kg)	0.20a (0.03)	0.23a (0.03)	0.19a (0.03)
Cage (EUR/kg)	0.08a (0.01)	0.07a (0.04)	0.10b (0.02)

^1^ heteroscedastic variable; a, b, ab means in a row within main effect (housing system).

**Table 4 animals-11-03040-t004:** Estimates of the effects of the housing system and the farmers within housing system in the surveyed farms (SD in brackets).

Housing System and Breeders	Technical Indexes			Economical Indexes							
	Walkable Cage Area (kg/m^2^)	Base Cage Area (kg/m^2^)	Total Shed Area (kg/m^2^)	Total VC (EUR/kg)	Feed ^1^ (EUR/Kg)	Drugs (EUR/kg)	Energy (EUR/kg)	Reproduction ^1^ (EUR/kg)	Total VF (EUR/kg)	Shed (EUR/kg)	Cage (EUR/kg)
Constant	207.24 *** (9.48)	207.24 *** (10.22)	115.34 *** (5.56)	1.33 *** (0.05)	1.03 *** (0.10)	0.14 *** (0.01)	0.08 *** (0.00)	0.08 *** (0.00)	0.27 *** (0.01)	0.19 *** (0.01)	0.08 *** (0.01)
BI	0 ^a^	0 ^a^	0 ^a^	0 ^a^	0 ^a^	0 ^a^	0 ^a^	0 ^a^	0 ^a^	0 ^a^	0 ^a^
DP	−23.26 (15.48)	−23.26 (16.69)	−15.61 (9.08)	−0.01 (0.09)	−0.02 (0.02)	−0.07 *** (0.02)	−0.01 ** (0.01)	0.09 *** (0.02)	0.6 ** (0.02)	0.03 * (0.02)	0.03 *** (0.01)
WRSA	−54.41 *** (17.73)	−11.38 (19.12)	3.01 (10.41)	−0.04 (0.10)	−0.09 *** (0.03)	−0.07 *** (0.02)	−0.05 *** (0.01)	0.18 ** (0.06)	0.03 (0.02)	0.00 (0.02)	0.03 ** (0.01)
BI X breeder A	0 ^a^	0 ^a^	0 ^a^	0 ^a^	0 ^a^	0 ^a^	0 ^a^	0 ^a^	0 ^a^	0 ^a^	0 ^a^
BI X breeder B	−76.55 *** (17.73)	−76.55 *** (19.12)	−25.01 ** (10.41)	−0.26 ** (0.10)	−0.10 *** (0.02)	−0.12 *** (0.02)	−0.05 *** (0.01)	0.01 (0.06)	0.07 ** (0.02)	−0.05** (0.02)	0.01 (0.01)
BI X breeder C	14.59 (17.73)	14.59 (19.12)	27.71 ** (10.41)	−0.02 (0.10)	0.06 ** (0.02)	−0.03 (0.02)	−0.05 *** (0.01)	−0.01 (0.05)	−0.05 * (0.02)	−0.04 * (0.02)	−0.01 (0.01)
DP X breeder D	0 ^a^	0 ^a^	0 ^a^	0 ^a^	0 ^a^	0 ^a^	0 ^a^	0 ^a^	0 ^a^	0 ^a^	0 ^a^
DP X breeder E	22.47 (16.19)	22.47 (17.45)	0.92 (9.50)	0.06 (0.09)	0.24 *** (0.07)	−0.02 (0.02)	−0.01 * (0.01)	−0.15 *** (0.02)	−0.07 (0.02)	−0.00 (0.02)	−0.07 *** (0.01)
WRSA X breeder C	0 ^a^	0 ^a^	0 ^a^	0 ^a^	0 ^a^	0 ^a^	0 ^a^	0 ^a^	0 ^a^	0 ^a^	0 ^a^
WRSA X breeder F	15.76 (25.96)	−45.69 (27.98)	1.02 (15.23)	−0.10 (0.15)	0.14 *** (0.03)	−0.04 (0.03)	−0.01 (0.01)	−0.20 *** (0.06)	−0.02 (0.03)	−0.01 (0.03)	−0.01 (0.01)
Adjusted R Squared	0.602	0.519	0.523	0.221	0.579	0.704	0.921	0.511	0.556	0.492	0.829

^a^ This parameter is set to zero because it is redundant; ^1^ heteroscedastic variable; *** *p*-value < 0.01, ** *p*-value < 0.05, * *p*-value < 0.10.

## Data Availability

The data are not publicly available in accordance with consent provided by participants on the use of confidential data.

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
