# Peer review of "The Economics of Rabbit Farming: A Pilot Study on the Impact of Different Housing Systems"

_animals, 2021, doi:10.3390/ani11113040_

Round 1
Reviewer 1 Report
Review of paper by Mondin et al., The economics of rabbit farming: a pilot study on the impact of different housing systems (Manuscript number: animals-1366418)
The authors evaluated a very important topic in the manuscript, conducting economic analysis and comparison of three housing systems. Very few publications have been published in this area so far, so their work is commended. It contains a lot of valuable and useful data. Nevertheless, in several cases, the reviewer has a sense of lack.
In addition to housing conditions, many factors affect the cost and profitability of the rabbit farms. It is impossible to deal with all of these in a manuscript. However, some factors should definitely be considered. These include:
- Age of cages used.This could be important because bicellular cages have been used for longer time than WRSA cages.The profitability of a rabbit farm could have an affect on replacing an older technology by a newer one.
- According to the reported data, there were 5-8 rabbits in a DP cage and 7-9 rabbits in WRSA cage. According to them, there may have been a difference in the litter size at weaning of the hybrid used. This has an important impact on cost and profitability. This is supported by the fact that there were significant differences in the economical indices between the farms within each housing system.
- In order to analyze this at least superficially, more information would be needed. E.g., there were significant differences between housing systems in weight of slaughtered rabbits (2.81, 2.71 and 2.66 kg). Were the rabbits sold at the same age or at a different age? This also affects costs. The same issue is reproductive rhythm or mortality, especially during fattening period, etc.
It would be important to show what was similar between each housing system or rabbit farm and how they differed. This can make the similarities and differences between the housing systems more objective, and it can explain what caused the differences between the rabbit farms.
My request is not an extension of the evaluation, but a more thorough analysis of identities and differences between housing systems and farms.
Author Response
We acknowledge the anonymous reviewer for the useful and precious suggestions that enabled us to greatly improve the quality of our manuscript. Accordingly, we completely revised the original paper just following your suggestions. Text modifies are highlighted with tracked changes.
Reviewer 1: The authors evaluated a very important topic in the manuscript, conducting economic analysis and comparison of three housing systems. Very few publications have been published in this area so far, so their work is commended. It contains a lot of valuable and useful data. Nevertheless, in several cases, the reviewer has a sense of lack. In addition to housing conditions, many factors affect the cost and profitability of the rabbit farms. It is impossible to deal with all of these in a manuscript. However, some factors should definitely be considered.
Authors: Thanks for your comment. Indeed, the authors are fully aware of the great deal of variability that characterizes a rabbit farm compared to other animal production systems (e.g. poultry production) as discussed also by EFSA (2020). Based on the suggestions of the reviewer, we added as much as possible evidence to the other factors affecting costs and profitability as detailed in next comments.
Reviewer 1: In addition to housing conditions, many factors affect the cost and profitability of the rabbit farms. It is impossible to deal with all of these in a manuscript. However, some factors should definitely be considered. These include:
- Age of cages used. This could be important because bicellular cages have been used for longer time than WRSA cages. The profitability of a rabbit farm could have an affect on replacing an older technology by a newer one.
Authors: thanks for your comment. Indeed, the age of the housing systems has been taken into account in the calculation since as mentioned in the manuscript “Fixed costs were determined as depreciation costs considering a depreciation time of 20 years for the structures (shed plus services) and 10 years for the cages” (now lines 159-161).
The depreciation cost of investments has been calculated starting from a current cost for investment, and was independent from the age of cages in the farm: see lines from 151 to 153 (“ The structures (shed, plants and cages) were considered as new investments and FC were estimated from their value based on commercial proposals.....”). In this way we eliminated any bias coming from different age of cages among different farms, excluding the cases of complete depreciation of investments for old structures that would distort our fixed cost estimation.
Surely, we agree with the reviewer that the replacement of a cage type with another type can improve the farm results because of the different technology. This effect is something that we intentionally assessed in the comparison of fixed costs indexes among housing systems. However, this improvements is dependant not only on the cage itself but also on the fact that renewal of cages is usually associated to renewal/improvement of the environmental control systems in the farm besides 1) a sanitary vacuum 2) a full change of the stock of reproducing does, which are usually associated to less sanitary problems and better productive results compared to previous production cycles. These issues have been outlined now on lines 309-314 (“Indeed, the drug costs were higher in farms adopting BI with relevant differences among farmers which were associated with differences in management, besides the age of the farms (lower in farms with WRSA cages compared to the other housing sys-tems). In fact, farms using WRSA cages underwent a full and complete all in-all out for renovation and renewed their reproducing does flock, which usually imply better health of animals”).
Reviewer 1:- According to the reported data, there were 5-8 rabbits in a DP cage and 7-9 rabbits in WRSA cage. According to them, there may have been a difference in the litter size at weaning of the hybrid used. This has an important impact on cost and profitability. This is supported by the fact that there were significant differences in the economical indices between the farms within each housing system.
Authors: thanks for your comment. Indeed data reported in table 1 for the line “expected” refer to the description of EFSA of the systems (as specified in the footnote). However, these differences in the number of rabbits per cage are not related to the litter size/hybrid, but are strictly dependant on the cage size. The WRSA cage is higher (more available surface included the platform) compared to DP for which farmers can put a higher number of animals in WRSA cages compared to DP.
Reviewer 1:- In order to analyze this at least superficially, more information would be needed. E.g., there were significant differences between housing systems in weight of slaughtered rabbits (2.81, 2.71 and 2.66 kg). Were the rabbits sold at the same age or at a different age? This also affects costs. The same issue is reproductive rhythm or mortality, especially during fattening period, etc.
Authors: we fully agree with the reviewer that slaughtering age, reproductive rhythm and mortality affect the costs. However, not all data were available from the farmers who also reported changes within the same farm from one observation to the other (one year to the other). The weight of the slaughtered rabbits was the only reliable and measurable data it was obtained. However, all the variability due to the factors mentioned by the Reviewer (besides others not mentioned) is covered when we compare production between the different housing systems based on kg of live weight produced during one year and we use the feeding/drugs costs etc. covered during the same year.
Thus, a sentence was added on now lines 267-270 to take into account the issues raised by the reviewer (“Within the different production systems, farm global results and, thus, costs and revenues change with the reproductive efficiency of the does, the rate at which meat rabbits reach slaughtering weight at different ages, as well as the morbidity and mortality of the different categories of rabbits.”).
Also the subsequent paragraph (now lines 271-275) was modified to take into account the reviewer’s observation (On the other hand, to the best of our knowledge, this is the first study providing data about how the different housing systems can affect the economic sustainability of farmers in a context characterized by high variability of all other production factors which is covered by the evaluation of the live weight produced and the variable costs incurred during one year.).
Reviewer 1: It would be important to show what was similar between each housing system or rabbit farm and how they differed. This can make the similarities and differences between the housing systems more objective, and it can explain what caused the differences between the rabbit farms.
Authors: the common feature was the “housing system” whereas all other issues (genotype, feeding plans and feed producers, slaughter age, reproduction management, farmer, etc. ) largely differed from one farm to another. The sentence of now lines 271-275 as modified following the previous comment should cover/answer to the reviewer’s observation (On the other hand, to the best of our knowledge, this is the first study providing data about how the different housing systems can affect the economic sustainability of farmers in a context characterized by high variability of all other production factors which is covered by the evaluation of the live weight produced and the variable costs incurred during one year.).
Reviewer 1: My request is not an extension of the evaluation, but a more thorough analysis of identities and differences between housing systems and farms.
Authors: thanks a lot. We fully understand your position and we tried to focus more on the points raised by the reviewer as detailed in the previous comments. The aim was to catch costs in systems having in common the housing systems, whereas other aspects/factors are different from one farm to another and can largely change within the same farm during one year or between years (e.g. slaughtering age/weight per slaughtered flock depending on market request or animal flock health, etc.; genetics within the same farm as for semen used in AI).
Reviewer 2 Report
This is an interesting manuscript that examines that overall costs associated with using conventional vs enriched caging for housing meat rabbits. The conclusions provide important information for moving welfare standards forward for farmed rabbits in the EU.
Pg 2 – throughout – please correct m2 to m2 in text
Pg 2, Line 83 – change ‘mine’ to ‘undermine’
Pg 2, Line 84 and 91 – remove ‘thus’ at the beginning of both sentences – unnecessary.
Pg 3, line 100 – minor grammar, please reword. Suggest: Face to face interviews were conducted on 6 farms between February and April 2019 and collected information over the 5-year period between 2014-2018 regarding type of rabbit housing, productivity and costs.
Pg 3, Table 1 – please define ‘n’ in column 1 of table 1 as well as in Table 3
Pg 3, line 121 – spelling – ‘feces’ not ‘faces’
Pg 4, table 2 – please fix alignment of text in middle column
Pg 5 – please use a consistent method when reporting p values – in text sometimes indicated by just a ‘P’ and sometimes by ‘p-value’
Pg 6, Table 4 – unclear what ‘intercept’ refers to – please clarify
Author Response
We acknowledge the anonymous reviewer for the useful and precious suggestions that enabled us to greatly improve the quality of our manuscript. Accordingly, we completely revised the original paper just following your suggestions. Text changes are highlighted with tracked changes.
Reviewer 2: This is an interesting manuscript that examines that overall costs associated with using conventional vs enriched caging for housing meat rabbits. The conclusions provide important information for moving welfare standards forward for farmed rabbits in the EU.
Authors: Thanks for your comment and for the following suggestions.
Reviewer 2: Pg 2 – throughout – please correct m2 to m2 in text
Authors: The text was modified along the whole manuscript as requested.
Reviewer 2: Pg 2, Line 83 – change ‘mine’ to ‘undermine’
Authors: The text was modified as requested (now line 93).
Reviewer 2: Pg 2, Line 84 and 91 – remove ‘thus’ at the beginning of both sentences – unnecessary.
Authors: The text was modified as requested (now lines 95 and 107).
Reviewer 2: Pg 3, line 100 – minor grammar, please reword. Suggest: Face to face interviews were conducted on 6 farms between February and April 2019 and collected information over the 5-year period between 2014-2018 regarding type of rabbit housing, productivity and costs.
Authors: The text was modified as requested (now lines 118-120).
Reviewer 2: Pg 3, Table 1 – please define ‘n’ in column 1 of table 1 as well as in Table 3
Authors: The text was modified as requested (see Table 1 and Table 3).
Reviewer 2: Pg 3, line 121 – spelling – ‘feces’ not ‘faces’
Authors: The text was modified as requested (now line 148).
Reviewer 2: Pg 4, table 2 – please fix alignment of text in middle column
Authors: The text was aligned as requested (see Table 2).
Reviewer 2: Pg 5 – please use a consistent method when reporting p values – in text sometimes indicated by just a ‘P’ and sometimes by ‘p-value’
Authors: The text was modified as requested and p-value was used along the whole manuscript.
Reviewer 2: Pg 6, Table 4 – unclear what ‘intercept’ refers to – please clarify
Authors: the term has been modified to “constant”. It refers to the constant of the regression.
Reviewer 3 Report
Simple summary: it is good and clear.
Keywords: the keywords repeat title words, and it is not good. The keywords must define the paper, but they need to be different from the title. So, I suggest changing all of them.
Introduction: The introduction provides a good background but I believe that more information about economics analyses should be added.
Materials and Methods
6 farms were used in the experiment. I didn't understand what the 18 annual observations mean. What was observed 18 times? What were the 18 yearly observations? Because physical structure did not change over the experimental period.
Besides, animal productive indexes used were annual.
Furthermore 9+7+3 = 19 and not 18. (???)
Using data 9, 7, and 3 it is an unbalanced plot problem. In this case, the statistical analysis adopted is not correct. It was preferable to use 3, 3, and 3 and eliminate this problem.
The location of properties is important information and must be in the text. The cost of production and the sale price of animals may vary depending on the region where the farms are located.
Line 110 - not to consider labour costs is a mistake. The family farm also has its price and not to consider this is to mask production costs.
Was the feed used in the 3 properties the same? Did it have the same cost and the same formulation? Different types of feed have different types of use by animals. The individual feed cost should be taken into account to get an idea of ​​the feed cost per animal/farm. In other words, a farm could use a more expensive feed and with worse feed conversion. The results would be masked. The same happens with drugs. Were the same drugs given to animals? At the same cost?
How was possible to compare feed consumption without explaining the sex, age, number of rabbits? More information needs to be added.
The paper aims to analyze the best kind of shed. What do FC (E/kg) and Drugs (E/Kg) suffer interference from the physical structure of the shed? This should be better explained in the work discussion.
Lines 114-115 “energy costs (these costs are assumed to be those more affected by the housing system and easy to analytically determine)”. And how was it determined? Explain it.
Table 3 – The discussion should start with “Slaughter weight (kg)”. This is important information. First of all, were all animals slaughtered at the same age? What was the percentage of each sex? Sex and age affect body weight.
Based on Table 3, there was no difference between “slaughter weights” between the 3 facilities. According to this table if “Total VC” and “Total FC” (if these calculations were done correctly) there was no difference between the 3 installations. So, for the producer, it makes no economic difference between which kind of installation to use. And this is the conclusion that this work should reach.
Obviously, other factors should have been taken into account by the authors, such as how many days until slaughter, age of animals, sex, feed conversion and etc. These factors will affect costs and productivity. In my opinion, this information is missing from the text and may affect the results.
Table 4 – I didn't understand this table. Its objective should be better explained. Furthermore, in the text it was not mentioned what farms A, B, C, D, and E mean.
Discussion
“Direct comparison with previously published results was not possible and some data retrieved from the field were used for comparison.” Ok. But there is a lot of information that must be discussed. For example: is the space on cages in accordance with animal welfare? which farm is most economically sustainable? Why? Compare it with different studies.
The discussion must improve and the authors need to find a conclusion for the manuscript. The authors need to discuss the results based on a point of view (economic, welfare, sustainability) and answer which kind of farm is better.
Conclusion
The manuscript needs to find a conclusion for the presented data.
Author Response
We acknowledge the anonymous reviewer for the useful and precious suggestions that enabled us to greatly improve the quality of our manuscript. Accordingly, we completely revised the original paper just following your suggestions. Text changes are highlighted with tracked changes.
Simple summary: it is good and clear.
Keywords: the keywords repeat title words, and it is not good. The keywords must define the paper, but they need to be different from the title. So, I suggest changing all of them.
Authors: The keywords were modified as follows: “bicellular cage; dual purpose cage; enriched cage, production costs; animal welfare” (now lines 38-39).
Introduction: The introduction provides a good background but I believe that more information about economics analyses should be added.
Authors: Thanks for your comment. A sentence was added about the possible approaches for an economic analysis at the farm level, outlining strength and weakness points and the reasons for the choice made in our study. In detail, the following sentence was added: “The literature cited above has addressed the issue of cost analysis by adopting the approach of the typical farm. This method does not require the collection of data at the farm-level for a significant number of farmers. As such, they are a considerably lower cost option relative to farm surveys, but they do not generally ensure the representativeness of the results” (now lines 102-106).
The following sentence has been added at the end of the introduction to highlight the innovative contribution of the paper “To the best of our knowledge, this is the first attempt to carry out a statistical analysis that compares, on the basis of the main cost elements observable at farm-level, these three housing systems for rabbit farms.” (lines 112-114)
Materials and Methods
6 farms were used in the experiment. I didn't understand what the 18 annual observations mean. What was observed 18 times? What were the 18 yearly observations? Because physical structure did not change over the experimental period.
Besides, animal productive indexes used were annual.
Furthermore 9+7+3 = 19 and not 18. (???)
Using data 9, 7, and 3 it is an unbalanced plot problem. In this case, the statistical analysis adopted is not correct. It was preferable to use 3, 3, and 3 and eliminate this problem.
Authors: We agree with the reviewer that a balanced number of observations per farm would have been better, but some farms did not collect all the data during the year, others were new and with fewer data. Thus, we choose to maintain in the analyses all the available data to increase the size of our data set.
The following sentence was added to clarify this issue “A total number of 19 observations (i.e. data referring to the yearly production) were collected from six farms (identified as A, B, C, D, E, F). One of them (farm C) changed the housing system in 2017, from bicellular to WRSA. A different number of observations was available per farm, for which an unbalanced dataset was used with 9 observations from farms with bicellular cages (BI) (9 observations x 3 farms); 7 observations from farms with conventional dual-purpose cages (DP) (7 observations x 2 farms); 3 observations from farms with structurally enriched cages (WRSA) (3 observations x 2 farms).” (now lines 123-130)
Reviewer 3: The location of properties is important information and must be in the text. The cost of production and the sale price of animals may vary depending on the region where the farms are located.
Authors: thanks for your comment. The requested information has been added on now lines 120-121 (Farms were located in the Regions of Veneto and Friuli Venezia Giulia, North-East Italy.)
Reviewer 3: Line 110 - not to consider labour costs is a mistake. The family farm also has its price and not to consider this is to mask production costs.
Authors: The authors are fully aware of this limit of the study. The aim of the research is not to fully assess production costs of rabbit farms, but to shed more light on the economic impact of different housing systems. The issue was already specified in the original manuscript (now lines 144-147) (“Since farmers in the sample were usually organised as a family farm with a diversified production, the labour costs associated with rabbit breeding was difficult to determine analytically and, thus, labour costs were not considered in this study”). Nevertheless, this issue has been now pointed out also in discussion (now lines 323-325).
Reviewer 3: Was the feed used in the 3 properties the same? Did it have the same cost and the same formulation? Different types of feed have different types of use by animals. The individual feed cost should be taken into account to get an idea of the feed cost per animal/farm. In other words, a farm could use a more expensive feed and with worse feed conversion. The results would be masked. The same happens with drugs. Were the same drugs given to animals? At the same cost?
Authors: Thanks for the comment. We are aware of this issue as it was partially covered in the original manuscript as specified at the end of this paragraph. Nevertheless, under commercial conditions, in the case of rabbit production, no standardization exists. Feeds can be produced by different private companies, using a large variety of raw materials and different chemical compositions between farms but also within the same farm. Please, note that a minimum of 3 diets is used for fattening rabbits and a minimum of 2 diets is used for reproducing does, plus a specific feed for replacement does and not pregnant does. Thus, it is correct to point out that feeding has a key role on productive efficiency, that its price can vary together with feed conversion, animal health and so on, but the only measures that can be used in our analysis are the total cost and the total quantity of all the feed bought along one year.
To take into account the reviewer’s comment, the sentence on now lines 298-301 has been slightly modified (Indeed, feed costs can largely vary according to all factors affecting feed intakes, such as diet composition (especially dietary energy and fibre contents) and feeding regime (restricted or ad libitum feeding), besides the health of the animals on a farm [27] and, least but not last, the feed company)
The absence of standardization works also for drugs. A high variability of commercial products were used in the recruited farms depending on the health issues (respiratory or enteric) and the rabbit category (does, growing rabbits). It is not possible to determine whether the same product was purchased at the same price in different farms.
Reviewer 3: How was possible to compare feed consumption without explaining the sex, age, number of rabbits? More information needs to be added.
Authors: rabbit farms all work with a closed cycle in which reproducing does are kept together with their litters. Rabbits have very limited sexual dimorphisms for which at commercial slaughtering age no relevant differences between males and females can be found and rabbits are kept and reared in mixed groups. Thus, for the economical analysis of the farm, the feed quantity that the farm use can vary with the number of reproducing does which represent the size of the farm. If data/costs are expressed per kg of live weight produced, all issues related to the farm size are taken into account since more does produce more growing rabbits, i.e. more meat.
Reviewer 3: The paper aims to analyze the best kind of shed. What do FC (E/kg) and Drugs (E/Kg) suffer interference from the physical structure of the shed? This should be better explained in the work discussion.
Authors: The comment is not fully clear to us. The paper aims to analyse costs per unit of live weight of meat rabbit produced in one year in farms using different types of housing systems (enclosures-cages/parks) for rabbits whereas the shed characteristics are similar because housing/enclosures (conventional, WRSA, parks) are all kept in closed shed made up of different materials (concrete, etc.) and using a different control of internal environmental conditions (by cooling for example).
Reviewer 3: Lines 114-115 “energy costs (these costs are assumed to be those more affected by the housing system and easy to analytically determine)”. And how was it determined? Explain it.
Authors: A sentence was added to clarify this issue, in detail “All costs incurred were obtained based on the bills of each year of observation per farm.” (now lines 143-144).
Reviewer 3: Table 3 – The discussion should start with “Slaughter weight (kg)”. This is important information. First of all, were all animals slaughtered at the same age? What was the percentage of each sex? Sex and age affect body weight.
Authors: As for slaughtering age/weight, it is true that obtaining higher weight at earlier age with lower mortality is better for costs. However, slaughter weight/age are not fixed traits in the same farm, they can change within the same farm along one production cycle (year) depending on market request, selling price, and special management needs of the farm.
As for sex, rabbits have very limited sexual dimorphisms for which at commercial slaughtering age no relevant differences between males and females can be found and rabbits are kept and reared in mixed groups. No control about the sex of slaughtered rabbits is performed in rabbit farms.
Reviewer 3: Based on Table 3, there was no difference between “slaughter weights” between the 3 facilities. According to this table if “Total VC” and “Total FC” (if these calculations were done correctly) there was no difference between the 3 installations. So, for the producer, it makes no economic difference between which kind of installation to use. And this is the conclusion that this work should reach.
Authors: That’s true. There’s no difference among the three housing systems for total VC and total FC as largely discussed in the original manuscript (now lines 221 and 232). A sentence was added to be more clear in conclusions on now lines 318-322 (“In conclusion, based on these preliminary results, since total FC or VC did not change among housing systems, whereas drug costs were lower in WRSA cages compared to BI cages, we can state that in a perspective of evolution towards good welfare practices in rabbit farming, the adoption of structurally enriched cages could be economically sustainable provided that market prices remain acceptable”).
Reviewer 3: Obviously, other factors should have been taken into account by the authors, such as how many days until slaughter, age of animals, sex, feed conversion and etc. These factors will affect costs and productivity. In my opinion, this information is missing from the text and may affect the results.
Authors: Thanks for the comments. Rabbit farms all work with a closed cycle in which reproducing does are kept together with their litters. Rabbits have very limited sexual dimorphisms for which at commercial slaughtering age no relevant differences between males and females can be found and rabbits are kept and reared in mixed groups. Thus, for the economical analysis of the farm, the feed quantity that the farm use can vary with the number of reproducing does which represent the size of the farm. If data/costs are expressed per kg of live weight produced, all issues related to the farm size are taken into account since more does produces more growing rabbits, i.e. more meat.
Based also on comments of previous reviewers, we added some more comments to take into account these factors of variability (now lines 267-270; 273-275).
Table 4 – I didn't understand this table. Its objective should be better explained. Furthermore, in the text it was not mentioned what farms A, B, C, D, and E mean.
Authors: The meaning of A, B, C...is described at the beginning of paragraph 3.2. It identifies in an anonymous way the farms involved in the survey. The identification of the farms has been added also in lines 124-125. Table 4 presents a list of regressions of different cost indexes over different housing systems (BI, DP, WRSA), including the breeders effect.
Discussion
“Direct comparison with previously published results was not possible and some data retrieved from the field were used for comparison.” Ok. But there is a lot of information that must be discussed. For example: is the space on cages in accordance with animal welfare? which farm is most economically sustainable? Why? Compare it with different studies.
Authors: We specified now in the mentioned sentence that we were speaking about “about the economics of rabbit farms” (now line 238-239). The present work was not addressed to discuss about animal welfare. Nevertheless, as requested by the reviewer, in introduction we added the following sentence: “According to EFSA [8, 17], welfare and health of reproducing does is likely lower in conventional cages compared to other systems (among which enriched cages and parks). As for growing rabbits, welfare is likely/highly likely lower in conventional cages (BI and DP cages) and higher in indoor parks, whereas no distinction can be made among the WRSA cages and niche systems (litter park, outdoor systems, organic systems) [8].” (now lines 78-83).
As for economic sustainability, conclusions were slightly modified to give evidence that all farms were economically sustainable “In conclusion, based on these preliminary results, since total FC or VC did not change among housing systems, whereas drug costs were lower in WRSA cages compared to BI cages, we can state that in a perspective of evolution towards good welfare practices in rabbit farming, the adoption of structurally enriched cages could be economically sustainable provided that market prices remain acceptable.“ (now lines 318-322).
Reviewer 3: The discussion must improve and the authors need to find a conclusion for the manuscript. The authors need to discuss the results based on a point of view (economic, welfare, sustainability) and answer which kind of farm is better.
Authors: The aim of the work was to evaluate only the economic aspects. Conclusions have been modified to be “conclusive” about this issue as follows: “In conclusion, based on these preliminary results, since total FC or VC did not change among housing systems, whereas drug costs were lower in WRSA cages compared to BI cages, we can state that in a perspective of evolution towards good welfare practices in rabbit farming, the adoption of structurally enriched cages could be economically sustainable provided that market prices remain acceptable.“ (now lines 318-322).
Conclusion
The manuscript needs to find a conclusion for the presented data.
Authors: Please, see the previous comment. Conclusions were more clearly given.
Reviewer 4 Report
Brief summary
In the present manuscript, three different rabbit housing conditions were compared in an economic point of view in detail. The manuscript is well-written and the study is well-designed.
Broad comments
The main strength of the manuscript is the selection of this hot topic and the method of data collection. Only one minor weakness is the interpretation of the drug cost-related results.
Minor comments and questions
Material and methods
Data collection
Page 3, line 100-102
The locations of the rabbit farms are missing.
The strain of rabbits is missing (e.g. New Zealand White, Hycole or hybrid of other strains).
Did all of the farmers use the same rabbit strain?
Data analysis
Page 4, line 165-167
The exact type of the statistical probe is missing, e.g. one-way ANOVA or two-way ANOVA.
Results
Page 5, lines 196-197:
What kind of drugs were used at those rabbit farms?
Did all of the farmers use the same drugs?
Did you consider the changes in the prices of those drugs over the years as well? If a rabbit farmer was able to buy a drug from a different distributor or at discount price, that information may have a great impact on your results. Therefore, the name of the drugs, their distributors and the price changes over the years should be added to the Results or Materials and Methods section.
Without these details, the drugs costs cannot be compared between different housing systems.
Moreover, it would be more informative, that the drug usage/animal was compared between the three different housing systems.
Author Response
We acknowledge the anonymous reviewer for the useful and precious suggestions that enabled us to greatly improve the quality of our manuscript. Accordingly, we completely revised the original paper just following your suggestions. Text changes are highlighted with tracked changes.
Brief summary
In the present manuscript, three different rabbit housing conditions were compared in an economic point of view in detail. The manuscript is well-written and the study is well-designed.
Authors: thanks for your comment.
Broad commentsnew
The main strength of the manuscript is the selection of this hot topic and the method of data collection. Only one minor weakness is the interpretation of the drug cost-related results.
Authors: thanks for your comment, we fully agree with the reviewer. We fully understand the possible limitations regarding the drug cost results. It is not possible to determine whether the same product was purchased at the same price in different farms. The availability of dose-based data on antimicrobial consumption (DDD/kg) is not covering the entire sample and period.
Minor comments and questions
Material and methods
Data collection
Page 3, line 100-102
The locations of the rabbit farms are missing.
Authors: thanks for your comment. The requested information has been added on now lines 120-121 (Farms were located in the Regions of Veneto and Friuli Venezia Giulia, North-East Italy.)
The strain of rabbits is missing (e.g. New Zealand White, Hycole or hybrid of other strains). Did all of the farmers use the same rabbit strain?
Authors: The information was not available . However, based on the field data and common practice for rabbit farmers, it is very likely that they were not using the same strains and that genotype changed within the same farm between different years.
Data analysis
Page 4, line 165-167
The exact type of the statistical probe is missing, e.g. one-way ANOVA or two-way ANOVA.
Authors: thanks for your comment. A sentence was added has been added on now lines:
- Line 195: “All the variables have a normal distribution and were analyzed with two-way ANOVA taking the fixed effect of housing system and management.
- line 199: “To assess the existence of a farm effect, the one-way ANOVA between-subjects was performed looking for statistically significant effects associated with the management (random effects)”
Results
Page 5, lines 196-197: What kind of drugs were used at those rabbit farms? Did all of the farmers use the same drugs? Did you consider the changes in the prices of those drugs over the years as well? If a rabbit farmer was able to buy a drug from a different distributor or at discount price, that information may have a great impact on your results. Therefore, the name of the drugs, their distributors and the price changes over the years should be added to the Results or Materials and Methods section. Without these details, the drugs costs cannot be compared between different housing systems. Moreover, it would be more informative, that the drug usage/animal was compared between the three different housing systems.
Authors: thanks for your comment. The information was not available and, under the conditions of rabbit farms, the expected standardization is impossible between years in the same farm and between farms (drugs are different and are bought from different purchasers). Even health problems for which rabbits are treated vary within the same farm and among farms (digestive, respiratory diseases, etc.).
As for differences between years in the costs of drugs, please note that costs for drugs (as for other input) all were discounted to 2018 - based on the price index…- to weigh the variation in the price of all costs. This issue was specified in the original manuscript (now lines 163-164; To allow a comparison among different years, all the monetary values were dis-counted to 2018 based on the price index of the goods purchased by farmers [28].).
Round 2
Reviewer 1 Report
Second review
L78-81: Are these opinions based on research findings or questionnaire responses? I do not think that the scientific results support them. For example, in deep litter there is a high risk of coccidiosis and death, in the case of free choice, rabbits do not prefer it.
Table 1: It would be very useful to write in Table 1 or in the text how old the cage types studied were. In less economically productive farms, there is usually not enough money for investment and cage replacement. This was only partially done.
Table 1: Why were the number of rabbits per cage different (5-8 or 7-9 growing rabbits / cage)? Was there a difference in number of kits per litter or in the death of the kits before weaning? This difference has a significant impact on profitability.
L209: p-value
Conclusions: It should be noted in the conclusions that it was not the WRSA cages alone were better, but also the replacement of traditional cages with WRSA cages at these farms, and the fact that hygiene conditions could be better in the newer cages, which contributed to less drug use.

Author Response
We acknowledge the anonymous reviewer for the useful suggestions that allowed us to further improve the quality of our manuscript.
Comments to Reviewer
L78-81: Are these opinions based on research findings or questionnaire responses? I do not think that the scientific results support them. For example, in deep litter there is a high risk of coccidiosis and death, in the case of free choice, rabbits do not prefer it.
Authors: as mentioned by the Reviewer, the health of rabbits in these systems is at a higher risk compared to the conventional ones. Nevertheless, the results of the EFSA opinion, based on via a 2-step expert knowledge elicitation process, were expressed in terms of the overall impact welfare score (including both health and behavioural welfare consequences).
All these issues have been now clarified in the introduction and the following sentence was added: “This Scientific Opinion [8] also compared the health and welfare of rabbits kept under different production systems by an overall impact score based on both health- and behavior-related welfare consequences and obtained through an Expert Knowledge Elicitation (EKE) process. According to the EKE results [8],…” (now lines 78-82)
Table 1: It would be very useful to write in Table 1 or in the text how old the cage types studied were. In less economically productive farms, there is usually not enough money for investment and cage replacement. This was only partially done.
Authors: Being a time series analysis it is not possible to have one value describing the age of cages. Therefore, the requested information was added in the text (now lines 134-136): “Regarding the age of the cages, it is possible to evaluate it for DP, installed on the 2 farms in 2010 and 2015 and for WRSA, installed on the 2 farms in 2017. BI cages were installed before 2014”. Additionally, we state that some years were missing because of a reduced number of production cycles (now line 128)
Table 1: Why were the number of rabbits per cage different (5-8 or 7-9 growing rabbits / cage)? Was there a difference in number of kits per litter or in the death of the kits before weaning? This difference has a significant impact on profitability.
Authors: As specified in Table 1 in the footnote “d”, the information in Table 1 referring to the number of rabbits per cage is “expected”. The information is included in the report of the European Commission as a general information and does not represent the number of rabbits/cage used in the tested study. In the present study, this information was not recorded since the systems were considered as a whole as commented in the first revision and evidenced in the text. Thus, to avoid any misunderstanding, this specific information was deleted from the text in table 1 since it has not been recorded in the study.
L209: p-value
Authors: The text was modified as requested (now line 220)
Conclusions: It should be noted in the conclusions that it was not the WRSA cages alone were better, but also the replacement of traditional cages with WRSA cages at these farms, and the fact that hygiene conditions could be better in the newer cages, which contributed to less drug use.
Authors: as requested, we have modified the conclusions to specify that “the replacement of traditional cages with structurally enriched cages, potentially associated with better hygiene conditions, could be economically sustainable provided that market prices remain acceptable.” (now lines 327-330).
Reviewer 3 Report
The authors made the suggested modifications and justified those that were not possible to be made.Author Response
The authors acknowledge the anonymous reviewer for the useful and precious suggestions in the revision process that enabled us to improve the quality of our manuscript.